# Risk Perception, Protective Behaviors, and General Anxiety during the Coronavirus Disease 2019 Pandemic among Affiliated Health Care Professionals in Taiwan: Comparisons with Frontline Health Care Professionals and the General Public

**DOI:** 10.3390/ijerph17249329

**Published:** 2020-12-13

**Authors:** Wei-Po Chou, Peng-Wei Wang, Shiou-Lan Chen, Yu-Ping Chang, Chia-Fen Wu, Wei-Hsin Lu, Cheng-Fang Yen

**Affiliations:** 1Graduate Institute of Medicine, College of Medicine, Kaohsiung Medical University, Kaohsiung 80708, Taiwan; webber1007@gmail.com (W.-P.C.); wistar.huang@gmail.com (P.-W.W.); shioulan@kmu.edu.tw (S.-L.C.); pino3015@hotmail.com (C.-F.W.); 2Department of Psychiatry, Kaohsiung Medical University Hospital, Kaohsiung 80708, Taiwan; 3School of Nursing, The State University of New York, University at Buffalo, New York, NY 14214-3079, USA; yc73@buffalo.edu; 4Department of Psychiatry, Ditmanson Medical Foundation Chia-Yi Christian Hospital, Chia-Yi City 60002, Taiwan

**Keywords:** affiliated health care worker, COVID-19, risk perception, protective behaviors, general anxiety

## Abstract

This study aimed to compare risk perception, information sources, adoption of protective behaviors against coronavirus disease 2019 (COVID-19), and levels of general anxiety among affiliated health care professionals, frontline health care professionals, and the general public in Taiwan. We recruited participants via a Facebook advertisement. We determined the risk perception, information sources, adoption of protective behaviors against COVID-19, and levels of general anxiety among 1954 respondents. In total, 269 affiliated healthcare workers, 371 frontline healthcare workers, and 1314 members of the general public were recruited into this study. The results indicated that both affiliated and frontline health care professionals had a higher level of risk perception of COVID-19, and more adopted protective behaviors against COVID-19 than the general public. No significant differences in risk perception or the adoption of protective behaviors were identified between affiliated, and frontline, health care professionals. Affiliated health care professionals had a lower level of general anxiety than the general public, whereas frontline health care professionals exhibited no significant difference in level of general anxiety compared with the general public or affiliated health care professionals. As important members of COVID-19 treatment teams, the need for psychological and educational support in affiliated health care professionals should receive attention.

## 1. Introduction

### 1.1. Well-Being of Health Care Professionals During Coronavirus Disease 2019

Coronavirus disease 2019 (COVID-19) is a highly contagious respiratory infectious disease that has spread rapidly worldwide [1]. As a novel viral infectious disease, COVID-19 has challenged modern medicine. Overall hospital mortality from COVID-19 is approximately 15% to 20%, but up to 40% among patients requiring admission to intensive care units [2]. Thus far, no vaccine for COVID-19 has been developed. Many countries lack the medical resources required to care for patients with COVID-19 [3]. The physical and psychological health of health care professionals is at greater risk during the COVID-19 pandemic because of an increased risk of infection, an excessive workload, unpreparedness, and emotional distress [4,5,6,7]. A review of 37 studies on the well-being of health care professionals during the COVID-19 pandemic indicated increased stress, anxiety, and depressive symptoms in health care professionals because of COVID-19; and worries regarding the possibility of infecting their families were particularly notable [7]. The well-being of health care professionals during the COVID-19 pandemic must be ensured to prevent the healthcare system from collapsing [7].

The well-being of frontline health care professionals involved in diagnosing or providing life-saving treatment to patients with COVID-19 has received considerable attention [8,9,10,11]. Although not directly involved in diagnosis and patient care, affiliated health care professionals have played important roles in medical teams during the COVID-19 pandemic [12]. The need for psychological and educational support in affiliated health care professionals may be different from that of frontline health care professionals. Understanding the cognitive, affective, and behavioral constructs of the health beliefs of affiliated health care professionals is essential for developing strategies to enhance their well-being during the COVID-19 pandemic [12].

### 1.2. Mental Health Problems in Frontline and Affiliated Health Care Professionals and the General Public

The differences in severity of mental health problems between affiliated and frontline health care professionals during the COVID-19 pandemic have been studied, with mixed results. Studies have reported that frontline health care professionals have greater anxiety [8,9,10,11], depression [8,9,11], insomnia [9], and distress [9] compared with non-frontline health care professionals. Nevertheless, no significant difference in depressive symptoms was noted between frontline and non-frontline health care professionals in another study [13]. A study conducted in China reported that frontline nurses had fewer symptoms of vicarious traumatization than did non-frontline nurses; however, no difference was observed between non-frontline nurses and members of the general public [14]. Moreover, a study conducted in Singapore found that the prevalence of anxiety was higher among nonmedical health care professionals (allied health professionals, pharmacists, technicians, administrators, clerical staff, and maintenance workers) than medical personnel (physicians and nurses); similarly, higher mean anxiety and stress subscale scores on the Depression, Anxiety, and Stress Scale, and higher total and intrusion, avoidance and hyperarousal subscale scores on the Impact of Events Scale, Revised were observed in nonmedical health care professionals [15]. A study in China during the COVID-19 outbreak (11–26 February 2020) comparing 1173 frontline, and 1173 age- and sex-matched non-frontline, medical workers found that frontline medical workers had higher rates of mental problems, anxiety symptoms, depressed mood, and insomnia than non-frontline medical workers; no significant difference was observed in terms of suicidal ideation, help-seeking, or treatment for mental problems [16].

Most studies were conducted in the early stages of the COVID-19 pandemic in China; whether mental health problems differ between frontline and affiliated health care professionals in other regions warrants further investigation. Moreover, few studies have compared mental health problems between frontline and affiliated health care professionals, and members of the general public [14,16,17].

### 1.3. Risk Perception, Information Sources, and Protective Behaviors in Frontline and Affiliated Health Care Professionals and the General Public

Risk perception refers to an individual’s subjective judgement regarding the characteristics and severity of a danger [18]. A study conducted in the United States revealed that people who perceived a greater risk of contracting COVID-19 were more likely to wash their hands and maintain social distancing to prevent becoming infected [19]; in China, risk perception was reported to be positively correlated with depressive states in patients with COVID-19 [20].

Dispensing timely and accurate information to the public and scientific community on COVID-19 is necessary for containing and curing the disease [21]. A study conducted in Jordan found that pharmacists who received COVID-19 information from the World Health Organization (WHO) website perceived the greatest risk of COVID-19 [22]. People who relied on unofficial sources of information, such as posts on social media platforms, tended to perceive a higher risk of COVID-19, but engaged in fewer protective behaviors [23]. Affiliated health care professionals were less likely to self-quarantine with their families, and washed their hands more frequently than frontline health care professionals [24]. Given that all health care professionals can help spread information regarding COVID-19 and adopt suitable protective behaviors, the risk perception, knowledge, and protective behaviors against COVID-19 of affiliated health care professionals warrant further investigation.

### 1.4. Study Aims

This study compared risk perception, information sources, adoption of protective behaviors against COVID-19, and levels of general anxiety among affiliated health care professionals, frontline health care professionals, and the general public in Taiwan. The first COVID-19 case in Taiwan was confirmed on 21 January 2020. During the period from 20 January to 24 February, the Taiwan Centers for Disease Control rapidly produced and implemented a list of more than 124 action items, including border control, case identification, quarantine of suspicious cases, proactive case finding, resource allocation, reassurance and education of the public while fighting misinformation, negotiation with other countries and regions, formulation of policies toward schools and childcare, and relief to businesses [25]. With proactive containment efforts and comprehensive contact tracing, the number of COVID-19 cases in Taiwan remained low, compared with other countries that had widespread outbreaks [26]. Therefore, there was no social lockdown in Taiwan. As of 27 November 2020, Taiwan had tested a total of 248,625 persons, showing 648 confirmed cases, of which only 55 were domestic. Seven patients died, and 556 people were released from hospital after testing negative three times subsequently [27]. However, the pandemic has profoundly affected the economy and unemployment rate in Taiwan [28]. We investigated the following hypotheses. The first hypothesis is that affiliated health care professionals perceive a lower risk of COVID-19, adopt fewer protective behaviors, have a lower level of general anxiety, and are less likely to receive COVID-19 information from formal education and medical professionals compared with frontline health care professionals. The second hypothesis is that affiliated health care professionals perceive a higher risk of COVID-19, adopt more protective behaviors, have higher levels of general anxiety, and are more likely to receive COVID-19 information from formal education and medical professionals compared with members of the general public.

## 2. Methods

### 2.1. Participants

The method of recruiting participants is comprehensively described elsewhere [29]. In brief, we used Facebook advertisements between 10 April 2020 and 23 April 2020. The inclusion criteria were as follows: aged 20 years or older, and residing in Taiwan. Participants accessed the study questionnaire website through Facebook advertisements. To ensure that health care professionals were recruited, we posted a link to the study questionnaire website on the Facebook pages of several health care worker associations in Taiwan. The Institutional Review Board of Kaohsiung Medical University Hospital approved this study (KMUHIRB-EXEMPT(I) 20200011), and waived the need for informed consent. We provided weblinks to related pages on the websites of the Taiwan Centers for Disease Control, Kaohsiung Medical University Hospital, and Medical College of National Cheng Kung University at the end of the online questionnaire for participants to access information related to COVID-19.

### 2.2. Measures

We reviewed literature studies on respiratory infectious diseases such as H1N1 influenza [30] and severe acute respiratory syndrome (SARS) [31]. Based on the results of the review, we developed the questionnaire for the study.

#### 2.2.1. Occupational Classification

Respondents were asked if they were frontline health care professionals (physicians and nurses who provide direct medical care to patients with COVID-19), affiliated health care professionals (those who work in medical units, and are in contact with, but not direct medical care providers for, patients with COVID-19), or members of the general public (those who are not health care professionals). Affiliated health care professionals were asked to provide their designation (e.g., non-frontline nurse, social worker, psychologist, occupational therapist, physical therapist, pharmacist, radiographer, nutritionist, or medical assistant).

#### 2.2.2. Sources of Information on COVID-19

We used the following questions to assess the frequency of the participants receiving COVID-19–related information from formal education (online or in person) or medical professionals: “How frequently do you receive COVID-19-related information from formal education (online or in person)/medical professionals?” Participants who rated their frequency of receiving information from a given source as “sometimes” or “frequently” were classified as having received COVID-19–related information from the source.

#### 2.2.3. Risk Perception of COVID-19

We transformed the five-item questionnaire developed by Liao et al. [30] for measuring risk perception of the H1N1 influenza, to measure risk perception of COVID-19. The questionnaire assessed the respondents’ level of worry regarding developing flu-like symptoms, worry regarding contracting COVID-19, and worry about the general situation in the world regarding COVID-19 as well as their perceived likelihood of contracting COVID-19, and perceived likelihood of contracting COVID-19 compared with others outside their family. The questions, response scales, and scores are listed in Appendix A. As the five items had various scoring methods, we divided the score of each question by its highest score, for example, the scores of items 1 and 2 were divided by 5, and the scores of items 4 and 5 were divided by 7, and we then summed the scores obtained (minimum: 0.786, maximum: 5). A higher total score indicates a greater risk perception.

#### 2.2.4. Adoption of Protective Behaviors Against COVID-19

We used the five questions developed by Liao et al. [30] to assess whether the participants adopted any of the five behaviors in the preceding week to protect themselves from contracting COVID-19: “In the past week, did you (1) avoid going to crowded places, (2) wash your hands more often, (3) wear a mask more often, (4) maintain good indoor ventilation, and (5) disinfect your household frequently?” Participants who responded “yes, due to COVID-19” were classified as practicing everyday COVID-19 protective behaviors (score 1); participants who responded “no” or “yes, but not due to COVID-19” were classified as not practicing everyday COVID-19 protective behaviors (score 0). The total numbers of adopted protective behaviors against COVID-19 were summed (minimum: 0, maximum: 5). The questions, response scales, and dichotomous scales used for statistical analysis are listed in Appendix A. A higher total score indicates the adoption of more protective behaviors against COVID-19.

#### 2.2.5. General Anxiety

We used the 10-item state anxiety scale from the state-trait anxiety inventory to assess the level of general anxiety experienced by the respondents in the preceding 7 days [30,31,32]. The questions, response scales, and scoring are listed in Appendix A. A higher total score indicates a higher level of general anxiety.

#### 2.2.6. Demographic Characteristics

We collected data on the sex and age of the respondents, and used the median age as the cutoff for age. Respondents ≤37 years formed the younger group, and those >37 years formed the older group.

### 2.3. Statistical Analysis

Data analysis was performed using SPSS statistical software (version 22.0; SPSS Inc., Chicago, IL, USA). The values of Cronbach’s α for the questionnaires assessing risk perception, adoption of protective behaviors, and general anxiety were 0.759, 0.722, and 0.921, respectively. The proportion of respondents in each of the three groups of occupations was compared using a chi-square test according to sex, age, and the possibility of receiving information regarding COVID-19 from formal education or medical professionals. We performed multivariate analysis of variance (MANOVA) to compare the levels of risk perception, adoption of protective behaviors, and general anxiety (independent variables) among the groups according to occupational classification (general public, affiliated health care professionals, and frontline health care professionals), sex (women and men), and age (younger and older). If occupational classification, sex, and age had significant effects on the differences in the dependent variables, then their interaction effects were further examined. The levels of risk perception, adoption of protective behaviors, and general anxiety were further compared between physicians and nurses who provided direct medical care to patients with COVID-19 using a *t*-test. A two-tailed *p* value of <0.05 was considered statistically significant.

## 3. Results

In total, the data of 269 affiliated healthcare workers, 371 frontline healthcare workers, and 1314 members of the general public were analyzed. Table 1 shows the results of the chi-square test comparing the proportions of respondents according to sex, age, and whether they received information regarding COVID-19 from formal education or medical professionals among the three occupation groups. Frontline health care professionals had a higher proportion of men than the other two groups, whereas both affiliated and frontline health care professionals had higher proportions of older respondents than the general public. Frontline health care professionals had the highest proportion of respondents receiving information regarding COVID-19 from formal education and medical professionals, followed by affiliated health care professionals and the general public.

The levels of risk perception, adoption of protective behaviors, and general anxiety (the dependent variables) were compared among the groups according to occupational classification, sex, and age using MANOVA. The results of significance in MANOVA are presented in Table 2, indicating that there were significant differences in the dependent variables among the groups of various occupational classifications, sex, and age (*p* < 0.001). However, the interactions of occupational classification with sex and age were not significantly associated with the differences in the dependent variables (*p* > 0.05). Thus, the effect of occupation was further examined.

Table 3 displays the results of MANOVA for comparing the levels of risk perception, adoption of protective behaviors, and general anxiety among the three occupation groups. Significant differences in the levels of risk perception (*p* < 0.001), adoption of protective behaviors (*p* = 0.012), and general anxiety (*p* = 0.014) were observed among the three occupation groups. The results of post hoc comparisons revealed that both affiliated and frontline health care professionals had a higher level of risk perception of COVID-19, and more adopted protective behaviors against COVID-19 than the general public. No significant differences in risk perception or adoption of protective behaviors were identified between affiliated and frontline health care professionals. Affiliated health care professionals had a lower level of general anxiety than the general public, whereas frontline health care professionals exhibited no significant differences in level of general anxiety compared with the general public or affiliated health care professionals.

Table 4 displays the results of the *t* test for comparing the levels of risk perception, adoption of protective behaviors, and general anxiety between physicians and nurses who provided direct medical care to patients with COVID-19. The results indicated that physicians adopted more protective behaviors against COVID-19 than nurses (*p* = 0.001), whereas nurses had a higher level of general anxiety than physicians (*p* = 0.048).

## 4. Discussion

### 4.1. Information on COVID-19

Consistent with our hypotheses, affiliated health care professionals were more likely than the general public, but less likely than frontline health care professionals, to receive information regarding COVID-19 from formal education and medical professionals. Taiwan was seriously affected by the 2002–2003 SARS epidemic, and had the third highest number of SARS cases globally, after China and Hong Kong [33]. On the basis of its experience of the SARS epidemic, Taiwan’s Ministry of Health and Welfare required all health care professionals to take formal online courses on epidemic prevention. Moreover, frontline health care professionals are trained to diagnose and treat patients with COVID-19 through online or in-person courses. Therefore, affiliated health care professionals have more opportunities to learn about COVID-19 from formal education and medical professionals than the general public. Owing to a lack of resources and limited ability to understand technical terms, the general public may not benefit from formal education on COVID-19. We recommend that both frontline and affiliated health care professionals seize any opportunity to deliver accurate information regarding COVID-19 to the public. However, 19.7% of affiliated, and 11.9% of frontline, health care professionals reported that they had not received COVID-19-related information from formal education before participating in this study. The results indicated that health policy authorities need to develop strategies to require health care professionals in any medical service units to promptly complete the formal online courses, and obtain the necessary knowledge and skills of COVID-19 related diagnosis and treatment.

### 4.2. Risk Perception and Protective Behaviors

Both affiliated and frontline health care professionals perceived a greater risk of COVID-19, and adopted more protective behaviors against COVID-19 compared with members of the general public. No significant differences in risk perception or adoption of protective behaviors were identified between affiliated and frontline health care professionals. Studies have revealed that, although perceiving a greater risk of contracting COVID-19 is positively correlated with the adoption of protective behaviors among members of the general public [19], some individuals have tended not to adopt protective behaviors, even after perceiving great risks of COVID-19, particularly those who rely on unofficial sources, such as gossip and news spread through friends on social media platforms [23]. Although affiliated health care professionals are not directly involved in treating patients with COVID-19, their access to information regarding COVID-19 from formal lessons and medical staff may imbue them with a realistic perception of the risk of COVID-19, and prompt them to adopt necessary protective behaviors.

### 4.3. General Anxiety

We discovered that affiliated health care professionals had lower general anxiety than the general public. No significant difference in general anxiety was found between affiliated and frontline health care professionals, which was in line with the results of the study by Liang et al. [13]. However, several studies have demonstrated that frontline health care professionals have greater anxiety compared with non-frontline health care professionals [8,9,10,11]. The differences in the results of the present and previous studies may have partially resulted from differences in timing, severity of COVID-19 pandemic, and methodological characteristics. For example, a follow-up study found that psychological distress increased more among healthcare, compared to non-healthcare, workers during the deterioration of the COVID-19 outbreak [34]. A review study on the psychiatric symptoms or morbidities associated with COVID-19 among infected patients, psychiatric patients, health care professionals, and non-health care professionals found that most of the studies were cross-sectional and varied in reported outcomes, measuring of outcomes, and statistical analysis [35]. Moreover, the COVID-19 pandemic has had severe adverse effects on employees, customers, supply chains, and financial markets [36]. Health care professionals may continue working during the pandemic, whereas a lot of the general public have lost jobs. Economic problems may be a potential reason to explain the difference in general anxiety between the general public and affiliated health care professionals.

Studies have reported the prevalence of general anxiety in the general public during the SARS outbreak [37,38] and the COVID-19 pandemic [39]. A lack of application of timely and efficient strategies against COVID-19 may increase levels of general anxiety [39]. Compared with the general public, affiliated health care professionals might be better equipped with timely and efficient strategies, because of existing medical knowledge and information received from formal lessons, which might further reduce their levels of general anxiety. The present study also found that nurses had a higher level of general anxiety than physicians. However, most of nurses participating into the present study were female (96.3%). Research found that female health care professionals were more likely to suffer from increased stress, anxiety, depressive symptoms, and insomnia than males [40]. Further study is needed to examine whether the difference in general anxiety between physicians and nurses exists beyond the gender effect.

### 4.4. Limitations

The present study had some limitations. First, although recruiting participants using Facebook advertisements can deliver large numbers of participants quickly [41], Facebook users may not be representative of the population. Second, the cross-sectional design of this study limited causal inference between the voluntary reduction of social interaction and perceived social support. Third, there might be factors, such as personality characteristics, which were not examined in the present study, that could account for both the voluntary reduction of social interaction and decreased social support. Research found that individuals who had psychiatric diseases were more likely to suffer from negative psychological impact during the COVID-19 epidemic than those who had no psychiatric diseases [42]. However, the present study did not survey participants’ psychiatric diagnoses before the COVID-19 pandemic. Fourth, participants completed the online questionnaire anonymously. Hence, it was impossible to verify the identity of the respondents and their responses in the survey. This needs to be clarified by a face-to-face interview study and by examining the test-retest reliability. Last, we surveyed participants’ general anxiety but did not survey their depression, stress, and posttraumatic stress disorder symptoms.

## 5. Conclusions

The present study demonstrated that affiliated health care professionals have a higher level of risk perception of COVID-19, adopt more protective behaviors against COVID-19, and have lower levels of general anxiety compared with the general public; moreover, affiliated health care professionals are more likely to receive information regarding COVID-19 from formal lessons and medical staff than are the general public. Although affiliated health care professionals are not responsible for making COVID-19 diagnoses or delivering life-saving treatment to patients infected with COVID-19, they have many opportunities to communicate with members of the general public and teach them coping skills. Thus, affiliated health care professionals may play a key role in preventing the spread of COVID-19.

## Figures and Tables

**Table 1 ijerph-17-09329-t001:** Comparisons Between the General Public, Affiliated Health Care Professionals, and Frontline Health Care Professionals in Taiwan, in Terms of Sex, Age, and Obtainment of Information Related to COVID-19: Chi-square test.

Variables	General Public (Group 1) (*n* = 1314)	Affiliated Health Care Professionals (Group 2) (*n* = 269)	Frontline Health Care Professionals (Group 3) (*n* = 371)	χ^2^	*p*	Post hoc Comparison
Men, *n* (%)	427 (32.5)	76 (28.3)	146 (39.4)	9.593	0.008	3 > 1, 2
Older age, *n* (%)	574 (43.7)	160 (59.5)	220 (59.3)	42.414	<0.001	2, 3 > 1
Receiving COVID-19-related information from formal education, *n* (%)	449 (34.2)	216 (80.3)	327 (88.1)	445.989	<0.001	3 > 2 > 1
Receiving COVID-19-related information from medical professionals, *n* (%)	429 (32.6)	221 (82.2)	339 (91.4)	523.331	<0.001	3 > 2 > 1

COVID-19: Coronavirus disease 2019.

**Table 2 ijerph-17-09329-t002:** Significance Tests of Multivariate Analysis of Variance Examining the Differences in Risk Perception, Adoption of Protective Behaviors, and General Anxiety Across Various Occupation Classifications, Sex, and Age.

Variables	Wilks’ Lambda	*F*	Hypothesis df	Error df	*p*
Occupation classification	0.973	8.880	6.000	3880.000	<0.001
Sex	0.986	8.92	3.000	1940.000	<0.001
Age	0.985	10.159	3.000	1940.000	<0.001
Occupation classification × Sex	0.998	0.739	6.000	3880.000	0.619
Occupation classification × Age	0.996	1.16	6.000	3880.000	0.322
Occupation classification × Sex × Age	0.997	1.132	6.000	3880.000	0.341

**Table 3 ijerph-17-09329-t003:** Comparisons of Risk Perception, Adoption of Protective Behaviors, and General Anxiety Among the General Public and Affiliated and Frontline Health Care Professionals: Multivariate Analysis of Variance.

Variables	General Public (Group 1) (*n* = 1314)	Affiliated Health Care Professionals (Group 2) (*n* = 269)	Frontline Health Care Professionals (Group 3) (*n* = 371)	Type III Sum of Squares	df	Mean Square	*F*	*p*	Post hoc Comparison
Risk perception of COVID-19, mean (SD)	2.866 (0.677)	2.982 (0.667)	3.072 (0.694)	12.317	2	6.159	13.635	<0.001	2, 3 > 1
Adoption of protective behaviors against COVID-19, mean (SD)	3.269 (1.568)	3.584 (1.447)	3.496 (1.488)	20.798	2	10.399	4.426	0.012	2, 3 > 1
General anxiety, mean (SD)	23.342 (6.723)	22.130 (6.252)	23.194 (6.163)	362.685	2	181.343	4.279	0.014	1 > 2

COVID-19: Coronavirus disease 2019; df: degree of freedom; SD: standard deviation.

**Table 4 ijerph-17-09329-t004:** Comparisons of Risk Perception, Adoption of Protective Behaviors, and Levels of General Anxiety between Physicians and Nurses Who Provided Direct Medical Care to Patients with COVID-19: *t* test.

Variables	Physicians (*n* = 267)	Nurses (*n* = 104)	*t*	*p*
Risk perception of COVID-19, mean (SD)	3.094 (0.698)	3.067 (0.701)	0.298	0.766
Adoption of protective behaviors against COVID-19, mean (SD)	3.730 (1.279)	3.134 (1.683)	3.262	0.001
General anxiety, mean (SD)	22.645 (6.291)	24.281 (6.387)	−1.990	0.048

COVID-19: Coronavirus disease 2019; SD: standard deviation.

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
