# Peer review of "Risk Perception, Protective Behaviors, and General Anxiety during the Coronavirus Disease 2019 Pandemic among Affiliated Health Care Professionals in Taiwan: Comparisons with Frontline Health Care Professionals and the General Public"

_ijerph, 2020, doi:10.3390/ijerph17249329_

Round 1

Reviewer 1 Report

Overall summary

The authors conducted a study to evaluate risk perception, information sources, adoption of protective behaviors against coronavirus disease 2019 (COVID-19), and levels of general anxiety among affiliated health care workers, frontline health care workers and general public in Taiwan.

General comments

Although the article is not too much ambitious, it is interesting, and it has great social and academic relevance. Thus, the following concerns should be addressed prior to this paper is considered for publication.

  • INTRODUCTION
  1. Line 39. The authors must review the references #1 and 2, because the mortality rate of COVID-19 is not included in them.
  2. Lines 49-56. The authors must include references for the following paragraph: “the well-being … COVID-19 pandemic”.
  3. Lines 70-71. The authors must include references for the following sentence: “moreover, few studies….general public”.

  • METHODS
  1. Please, include the language which was used in the questionnaire and explain how the questionnaire was designed.
  2. Lines 157-167. In the section “statistical analysis” it is necessary to include that the psychometric properties (ie, Cronbach coefficient) for the sections: i) risk perception of COVID-19, ii) adoption of protective behaviors against COVID-19 and iii) general anxiety of the questionnaire were studied.
  3. When a scale is used, it is very useful to include the minimum and the maximum scale values. Please, include this information in the main text for the subsections: 2.2.3. Risk perception of COVID-19; and 2.2.4. Adoption of protective behaviors against COVID-19.
  4. Supplementary Table 1: a) Please, review the item 3 of the section “risk perception of COVID-19”, it evaluates the level of concern towards human swine flu; b) Explain in the main text why the authors divided by 5 (item 1 and 2, risk perception of COVID-19), by 10 (item 3, risk perception of COVID-19)…the scores obtained, instead of adding up the scores obtained.

  • RESULTS
  1. Have the authors registered the profession of the frontline health care workers? It would be interesting to analyse if there are differences between nurses and physicians who provide direct medical care to patients with COVID-19.
  2. As you know, tables must be auto-explicative. So, in Table 1-3 legends should, at least, include the statistical test which was used for the comparison of the results. Likewise: a) the title of the Table 2 should be changed, it is not an explanatory title; b) the data of the three first columns are expressed as mean (SD), median (SD)??? I do not know what they exactly mean.

  • DISCUSSION

  1. In the discussion the authors must try to explain why the studies carried out to evaluate mental health problems between affiliated and frontline health care workers and the general public during the COVID-19 pandemic could have been mixed. Perhaps methodological differences can explain it.
  2. The prevalence of general anxiety could be related to economic problems (health care workers have could work during the pandemic, but a lot of general public has lost the job). Maybe, economic problems are another potential reason to explain the differences observed in relation to general anxiety between general public and affiliated health care workers.
  3. Although taking formal online courses on epidemic prevention is compulsory for health care workers in Taiwan (lines 208-209); according to data from table 1, the ~20% of the affiliated healthcare workers and the ~12% of the frontline healthcare workers did not receive COVID-19 related information from formal education. Have the authors any explanation?

  • REFERENCES
  1. References # 10-12: The years must be written in bold.
  2. The references #18, 19 and 22 are all in bold.

Author Response

INTRODUCTION

Comment 1

Line 39. The authors must review the references #1 and 2, because the mortality rate of COVID-19 is not included in them.

Response

  • Thank you for your reminding. We corrected this sentence as below. Please refer to line 39-42.

“Coronavirus disease 2019 (COVID-19) is a highly contagious respiratory infectious disease that has spread rapidly worldwide [1]. As a novel viral infectious disease, COVID-19 has challenged modern medicine. Overall hospital mortality from COVID-19 is approximately 15% to 20%, but up to 40% among patients requiring admission to the intensive care units [2].”

  • We also changed references 1 and 2 into two new ones as below. Please refer to line 357-361.

Reference 1. Science and Engineering at Johns Hopkins. Modeling 2019-nCoV. Available online:  https://systems.jhu.edu/research/public-health/ncov-model/(accessed on 26 November 2020)

Reference 2. Wiersinga, W.J.; Rhodes, A.; Cheng, A.C.; Peacock, S.J.; Prescott, H.C. Pathophysiology, transmission, diagnosis, and treatment of coronavirus disease 2019 (COVID-19): A review. JAMA. 2020, 324, 782-793. doi: 10.1001/jama.2020.12839.

Comment 2

Lines 49-56. The authors must include references for the following paragraph: “the well-being … COVID-19 pandemic”.

Response

We added references 8 to 12 into this paragraph. Please refer to line 53, 55 and 59.

Comment 3

Lines 70-71. The authors must include references for the following sentence: “moreover, few studies….general public”.

Response

We added references 14, 16 and 17 into this paragraph. Please refer to line 84.

METHODS

Comment 4

Please, include the language which was used in the questionnaire and explain how the questionnaire was designed.

Response

We added the questions used in the present study as below.

  • “How frequency do you receive COVID-19-related information from formal education (online or in person)/medical professionals?” Please refer to line 153-154.
  • If you were to develop flu-like symptoms tomorrow, would you worry? score: 1-5”; “In the past one week, have you ever worried about catching COVID-19? score: 1-5”; “Please rate the current level of your worry towards COVID-19. score: 1-10”; “How likely do you think it is that you will contract COVID-19 over the next 1 month? score: 1-7”; “What do you think are your chances of getting COVID-19 over the next 1 month compared to others outside your family? score: 1-7 Please refer to line 161-173.
  • In the past week, did you 1) avoid going to crowded places, 2) wash your hands more often, 3) wear a mask more often, 4) maintain good indoor ventilation, and 5) disinfecting household frequently?” Please refer to line 177-179.
  • “Please rate the level of your agreement on 10 statements: You feel rested/content/comfortable/relaxed/pleasant/ anxious/nervous/jittery/“high strung”/over-excited and rattled.” Please refer to line 188-190.
  1. We explained how we designed the questionnaire in the revised manuscript.
  • “We reviewed the literature studied on respiratory infectious diseases such as severe acute respiratory syndrome (SARS) and H1N1 influenza. Based on the results of review, we developed the questionnaire for the previous study.” Please refer to line 140-142.
  • “We transformed the 5-item questionnaire developed by Liao et al. [26] for measuring risk perception of the H1N1 influenza to measure risk perception of COVID-19.” Please refer to line 159-160.
  • “We used the 5 questions developed by Liao et al. [26] to assessed whether the participants adopted any of the five behaviors in the preceding week to protect themselves from contracting COVID-19…” Please refer to line 175-177.
  • “We used the 10-item state anxiety scale from the State-Trait Anxiety Inventory to assess the level of general anxiety…” Please refer to line 187-188.

Comment 5

Lines 157-167. In the section “statistical analysis” it is necessary to include that the psychometric properties (ie, Cronbach coefficient) for the sections: i) risk perception of COVID-19, ii) adoption of protective behaviors against COVID-19 and iii) general anxiety of the questionnaire were studied.

Response

We added Cronbach coefficients into “statistical analysis” as below. Please refer to line 199-200.

“The value of Cronbach’s α for the questionnaires assessing risk perception, adoption of protective behaviors, and general anxiety was 0.759, 0.722, and 0.921 in this study, respectively.”

Comment 6

When a scale is used, it is very useful to include the minimum and the maximum scale values. Please, include this information in the main text for the subsections: 2.2.3. Risk perception of COVID-19; and 2.2.4. Adoption of protective behaviors against COVID-19.

Response

We included the minimum and the maximum scale values as below into “2.2.3. Risk perception of COVID-19” and “2.2.4. Adoption of protective behaviors against COVID-19” in the revised manuscript.

  • minimum: 0.786, maximum: 5” Please refer to line 172-173.
  • “Participants who responded “yes, due to COVID-19” were classified as practicing everyday COVID-19 protective behaviors (score 1); participants who responded “no” or “yes, but not due to COVID-19” were classified as not practicing everyday COVID-19 protective behaviors (score 0). The total numbers of adopted protective behaviors against COVID-19 were summed (minimum: 0, maximum: 5). Please refer to line 179-183.

Comment 7

Supplementary Table 1: a) Please, review the item 3 of the section “risk perception of COVID-19”, it evaluates the level of concern towards human swine flu; b) Explain in the main text why the authors divided by 5 (item 1 and 2, risk perception of COVID-19), by 10 (item 3, risk perception of COVID-19)…the scores obtained, instead of adding up the scores obtained.

Response

  1. a) Thank you for your reminding. We replaced “human swine flu” by “COVID-19” in Supplementary Table 1.
  2. b) We added the explanations for the scoring methods of risk perception as below into the text. Please refer to line 170-173.

Because that the 5 items had various scoring methods, we divided the score of each question by its highest score, for example, the scores of items 1 and 2 divided by 5 and the scores of items 4 and 5 divided by 7 and then summed up the scores obtained (minimum: 0.786, maximum: 5).”

RESULTS

Comment 8

Have the authors registered the profession of the frontline health care workers? It would be interesting to analyze if there are differences between nurses and physicians who provide direct medical care to patients with COVID-19.

Response

Thank you for your suggestion. We compared the levels of risk perception, adoption of protective behaviors, and general anxiety were further compared between physicians and nurses who provide direct medical care to patients with COVID-19. We found that physicians adopted more protective behaviors against COVID-19 than nurses (p = 0.001), whereas nurses had a higher level of general anxiety than physicians (p = 0.048). We added them into the revised manuscript as below.

Methods, 2.3. Statistical Analysis: “The levels of risk perception, adoption of protective behaviors, and general anxiety were further compared between physicians and nurses who provide direct medical care to patients with COVID-19 using t test.” Please refer to line 20-211.

Results: Table 4 displays the results of t test for comparing the levels of risk perception, adoption of protective behaviors, and general anxiety between physicians and nurses. The results indicated that physicians adopted more protective behaviors against COVID-19 than nurses (p = 0.001), whereas nurses had a higher level of general anxiety than physicians (p = 0.048).” Please refer to line 254-262.

Discussion: The present study also found that nurses had a higher level of general anxiety than physicians. However, most of nurses participating into the present study were female (96.3%). Research found that female health care workers were more likely to suffer from increased stress, anxiety, depressive symptoms, insomnia than males [40]. Further study is needed to examine whether the difference in general anxiety between physicians and nurses exists beyond the gender effect. Please refer to line 319-324.

Comment 9

As you know, tables must be auto-explicative. So, in Table 1-3 legends should, at least, include the statistical test which was used for the comparison of the results. Likewise: a) the title of the Table 2 should be changed, it is not an explanatory title; b) the data of the three first columns are expressed as mean (SD), median (SD)??? I do not know what they exactly mean.

Response

  • Thank you for your comments. We revised the titles of Tables 1 to 3 as below to make them more auto-explicative.

Table 1. Comparisons Between the General Public, Affiliated Health Care Workers, and Frontline Health Care Workers in Taiwan in Terms of Sex, Age, and Obtainment of Information Related to COVID-19: Chi-square test.” Please refer to line 224-226.

“Table 2. Significance Tests of Multivariate Analysis of Variance Examining the Differences in Risk Perception, Adoption of Protective Behaviors, and General Anxiety Across Various Occupation Classification, Sex and Age.” Please refer to line 235-237.

“Table 3. Comparisons of Risk Perception, Adoption of Protective Behaviors, and General Anxiety Among the General Public and Affiliated and Frontline Health Care Workers: Multivariate Analysis of Variance.” Please refer to line 250-252.

  • We added “mean (SD)” into the first column of Table 3.

DISCUSSION

Comment 10

In the discussion the authors must try to explain why the studies carried out to evaluate mental health problems between affiliated and frontline health care workers and the general public during the COVID-19 pandemic could have been mixed. Perhaps methodological differences can explain it.

Response

Thank you for your suggestion. We added discussion on possible reasons accounting for the mixed results of studies as below. Please refer to line 298-308.

No significant difference in general anxiety was found between affiliated and frontline health care professionals, which was in line with the result of the study of Liang et al. [13]. However, several studies have demonstrated that frontline health care professionals have greater anxiety compared with nonfrontline health care professionals [8-11]. The differences in the results of previous and present studies may partially result from the differences in timing, severity of COVID-19 pandemic and methodological characteristics. For example, a follow-up study found that psychological distress increased more among healthcare compared to non-healthcare workers during the deterioration of COVID-19 outbreak [34]. A review study on psychiatric symptoms or morbidities associated with COVID-19 among infected patients, psychiatric patients, health care professionals and non-health care professionals found that most of the studies were cross-sectional and various in reported outcomes, measuring of outcomes and statistical analysis [35].”

Comment 11

The prevalence of general anxiety could be related to economic problems (health care workers have could work during the pandemic, but a lot of general public has lost the job). Maybe, economic problems are another potential reason to explain the differences observed in relation to general anxiety between general public and affiliated health care workers.

Response

Thank you for your reminding. We added it as one of possible reasons accounting for the difference in general anxiety between general public and affiliated health care workers. Please refer to line 308-315.

“Moreover, the COVID-19 pandemic has severe adverse effects on the employees, customers, supply chains and financial markets [36]. Health care professionals may continue working during the pandemic, whereas a lot of general public has lost the job. Economic problems may be a potential reason to explain the difference in general anxiety between general public and affiliated health care professionals.”

Comment 12

Although taking formal online courses on epidemic prevention is compulsory for health care workers in Taiwan (lines 208-209); according to data from table 1, the ~20% of the affiliated healthcare workers and the ~12% of the frontline healthcare workers did not receive COVID-19 related information from formal education. Have the authors any explanation?

Response

Thank you for your reminding. We added discussion on the result as below. Please refer to line 277-282.

However, 19.7% of affiliated and 11.9% of frontline health care workers reported that they did not receiving COVID-19-related information from formal education before participating this study. The result indicated that health policy authorities need to develop strategies to require health care workers in any medical service units to timely complete the formal online courses and obtain necessary knowledge and skills of COVID-19 related diagnosis and treatment.”

REFERENCES

Comment 13

  1. References # 10-12: The years must be written in bold.
  2. The references #18, 19 and 22 are all in bold.

Response

Thank you for your reminding. We corrected and renumbered them. Please refer to references 10, 11, 13, 21, 22 and 23.

Reviewer 2 Report

This is a paper on an interesting topic in the current times. It is focused on the comparison between risk perception, information sources and protective behaviours against COVID-19. The authors compared three well-differentiated groups: 1) affiliated healthcare workers, 2) frontline healthcare workers and 3) general public.

I consider it is of quality. However, before the publication of the paper, some minor changes should be made.

 In the introduction section, the authors reported that a study conducted in Singapore found that nonmedical workers showed higher anxiety and stress compared to individuals who were medical workers. I would add some more details on this study. I consider that this study is of relevance, and should be better described.

The pandemic situation in Taiwan should be better described in the introduction section. I consider that a better understanding of the specific situation of Taiwan would help the readers to comprehensively describe the results. The authors have briefly described it at the first paragraph of the discussion section.

Prevalence of previous psychiatric diagnoses were not taken into account in the statistical analyses. The authors should add this limitation at the end of the manuscript, at the limitations section.

One more potential limitation is that the authors did not evaluate depressive symptoms according to a validated and well-recognized scale.This should be also mentioned in the limitations section.

Author Response

Comment 1

In the introduction section, the authors reported that a study conducted in Singapore found that nonmedical workers showed higher anxiety and stress compared to individuals who were medical workers. I would add some more details on this study. I consider that this study is of relevance, and should be better described.

Response

Thank you for your suggestion. We described the results of the study conducted in Singapore more as below. Please refer to line 69-75.

“Moreover, a study conducted in Singapore found that the prevalence of anxiety was higher among nonmedical health care workers (allied health professionals, pharmacists, technicians, administrators, clerical staff, and maintenance workers) than medical personnel (physicians and nurses); similarly, higher mean anxiety and stress subscale scores on the Depression, Anxiety, and Stress Scales and higher total and intrusion, avoidance and hyperarousal subscale scores on the Impact of Events Scale–Revised were observed in nonmedical health care workers [15].”

Comment 2

The pandemic situation in Taiwan should be better described in the introduction section. I consider that a better understanding of the specific situation of Taiwan would help the readers to comprehensively describe the results. The authors have briefly described it at the first paragraph of the discussion section.

Response

We added more introduction for the pandemic situation in Taiwan as below. Please refer to line 106-118.

The first COVID-19 case in Taiwan was confirmed on 21 January 2020. During the period from January 20 to February 24, the Taiwan Centers for Disease Control rapidly produced and implemented a list of at least 124 action items including border control, case identification, quarantine of suspicious cases, proactive case finding, resource allocation, reassurance and education of the public while fighting misinformation, negotiation with other countries and regions, formulation of policies toward schools and childcare, and relief to businesses [25]. With proactive containment efforts and comprehensive contact tracing, the number of COVID-19 cases in Taiwan remained low, as compared with other countries that had widespread outbreaks [26]. Therefore, there was no social lockdown in Taiwan. As of 27 November 2020, Taiwan had tested a total of 248,625 persons showing 648 confirmed cases, of which only 55 were domestic. Seven patients died, and 556 people were released from hospital after testing negative three times subsequently [27]. However, the pandemic has profoundly affected the economy and unemployment rate in Taiwan [28].”

Comment 3

Prevalence of previous psychiatric diagnoses were not taken into account in the statistical analyses. The authors should add this limitation at the end of the manuscript, at the limitations section.

Response

Thank you for your suggestion. We added it as one of the limitations as below. Please refer to line 332-335.

“Research found that individuals who had psychiatric diseases were more likely to suffer from negative psychological impact during the COVID-19 epidemic than those who had no psychiatric diseases [42]. However, the present study did not survey participants’ psychiatric diagnoses before the COVID-19 pandemic.”

Comment 4

One more potential limitation is that the authors did not evaluate depressive symptoms according to a validated and well-recognized scale. This should be also mentioned in the limitations section.

Response

We added it as one of the limitations as below. Please refer to line 338-339.

“Last, we surveyed participants’ general anxiety but did not survey their depression, stress, and posttraumatic stress disorder symptoms.”

Reviewer 3 Report

Recommendation: Minor revision

In their paper, Chou et al assessed risk perception and protective behaviors against coronavirus disease 2019 (COVID-19) as well as anxiety level among health care professionals and the general population in Taiwan. Via Facebook advertisement, the authors recruited in total 1954 study participants from the public, 296 affiliated health care workers, who were not directly providing medical care, and 371 frontline health care workers, including physicians and nurses directly taking care of patients. Study participants were classified according to their occupational status in three groups:affiliated and frontline health care professionals as well as non-health care workers. Risk perception was assessed using a modified 5-item questionnaire developed previously to measure risk perception during the H1N1 influenza pandemic, whereas general anxiety was measured using the State-Trait Anxiety Inventory. The authors found no significant differences in the levels of risk perception and adoption of protective behaviors between the two groups of affiliated health care workers and frontline health care workers. Likewise, general anxiety scores did not differ between these two groups. In contrast, affiliated health care workers had a significantly lower level of general anxiety than the general public, while frontline health care workers exhibited no significant differences in their level of general anxiety. It is not surprising that affiliated health care workers more likely received information regarding COVID-19 from formal medical lessons and interactions with staff.

Overall, the paper is well written and the methodology is sound.

However, I do not understand the following sentence in the Results section: “Occupation, sex, and age were significantly associated with differences in the dependent variables (p < 0.001)”. How were these models created? Please give more details.

The authors should consider using the term “health care professionals” instead of “health care worker”. Please also give the meaning of (or examples for) “affiliated” and ”frontline” health care professionals earlier in the manuscript, either in the Abstract or the Introduction.

Reference 18, 19 and 22 are completely written in bold letters. Please change.

Author Response

Comment 1

I do not understand the following sentence in the Results section: “Occupation, sex, and age were significantly associated with differences in the dependent variables (p < 0.001)”. How were these models created? Please give more details.

Response

Thank you for your comment. We revised this sentence as below. Please refer to line 228-232.

The levels of risk perception, adoption of protective behaviors, and general anxiety (the dependent variables) were compared among the groups according to occupational classification, sex, and age using MANOVA. The results of significance in MANOVA are presented in Table 2, indicating that there were significant differences in the dependent variables among the groups of various occupational classification, sex, and age (p < 0.001).”

Comment 2

The authors should consider using the term “health care professionals” instead of “health care worker”. Please also give the meaning of (or examples for) “affiliated” and ”frontline” health care professionals earlier in the manuscript, either in the Abstract or the Introduction.

Response

Thank you for your suggestion. We used the term “health care professionals” instead of “health care worker” thorough the revised manuscript.

Comment 3

Reference 18, 19 and 22 are completely written in bold letters. Please change.

Response

Thank you for your reminding. We corrected and renumbered them. Please refer to references 10, 11, 13, 21, 22 and 23.

Round 2

Reviewer 1 Report

The manuscript has been significantly improved and the authors have addressed the changes which were previously suggested. However, as it is noted in detail below, there are a few concerns that should be addressed prior to this paper is considered for publication.

1.Comment 4:

a) It is not necessary to include the questions of the questionnaire in the manuscript because they are in the Table S1. It is enough if the authors include in the text the language they used. Sometimes, the questions included in supplementary materials are translations, but in this case it is not so.

b) On the other hand, I am sorry, but I don´t understand the sentence “Based on the results of review, we developed the questionnaire for the previous study” (lines 140-141). Which is the previous study?

2. Line 172. Review the sentence “The range of total A higher total score indicates a greater risk perception.”

3. Line 174. Change “to assessed” by “to assess”.

Author Response

Thank you for your comments. We revised the manuscript based on your comments.

Comment

  1. It is not necessary to include the questions of the questionnaire in the manuscript because they are in the Table S1. It is enough if the authors include in the text the language they used. Sometimes, the questions included in supplementary materials are translations, but in this case it is not so.

Response

We deleted the questions in the revised manuscript and kept them in the Table S1. Please refer to line 158-167 and 181-184.

Comment

  1. b) On the other hand, I am sorry, but I don´t understand the sentence “Based on the results of review, we developed the questionnaire for the previous study” (lines 140-141). Which is the previous study?

Response

We added the references into this sentence as below. Please refer to line 139-140.

“We reviewed the literature studied on respiratory infectious diseases such as H1N1 influenza [30] and severe acute respiratory syndrome (SARS) [31].”

Comment

  1. Line 172. Review the sentence “The range of total A higher total score indicates a greater risk perception.”

Response

We revised this sentence as below. Please refer to line 167.

“A higher total score indicates a greater risk perception.”

Comment

  1. Line 174. Change “to assessed” by “to assess”.

Response

We changed “to assessed” by “to assess”. Please refer to line 169.